Whole-brain radiation therapy plus simultaneous integrated boost for brain metastases from breast cancers

Zhang Hongyan
http://orcid.org/0000-0002-4655-3108 Wu Qiuji wuqiuji@126.com
Li Li
Wang Linwei
Zhong Yahua doctorzyh73@163.com
Department of Radiation and Medical Oncology, Hubei Key Laboratory of Tumor Biological Behaviors, Hubei Cancer Clinical Study Center, Zhongnan Hospital of Wuhan University , Wuhan, Hubei Province , China
Oliveira Sonia
Electronic publication date: 2024 Jul 12
Publication date: 2024
Volume: 12
Electronic Location ID: e17696
Received 2023 Nov 14; Accepted 2024 Jun 14
Copyright: © 2024 Zhang et al.
Copyright year: 2024
Copyright holder: Zhang et al.
License: This is an open access article distributed under the terms of the Creative Commons Attribution License, which permits unrestricted use, distribution, reproduction and adaptation in any medium and for any purpose provided that it is properly attributed. For attribution, the original author(s), title, publication source (PeerJ) and either DOI or URL of the article must be cited.
License URL: https://creativecommons.org/licenses/by/4.0/

Keywords: Breast cancer, Brain metastasis, WBRT, Simultaneous integrated boost, Outcome

Funding: Science and Technology Innovation Cultivation Fund for Clinical Research Project from Zhongnan Hospital of Wuhan University lcyf202213 This study was supported by the Science and Technology Innovation Cultivation Fund for Clinical Research Project from Zhongnan Hospital of Wuhan University (No. lcyf202213). The funders had no role in study design, data collection and analysis, decision to publish, or preparation of the manuscript.

==============================
Background

The effect of whole-brain radiation therapy (WBRT) plus simultaneous integrated boost (SIB) in brain metastasis from breast cancers has not been demonstrated.

Method

In this single-center retrospective study, we reviewed consecutive breast cancer patients who developed brain metastasis and were treated with hypofractionated radiation therapy plus WBRT using intensity-modulated radiation therapy (IMRT)-SIB approaches. We analyzed clinical outcomes, prognostic factors and patterns of treatment failure.

Result

A total of 27 patients were eligible for analysis. Four (14.8%) patients achieved clinical complete response and 14 (51.9%) had partial response of brain lesions. The other nine patients were not evaluated for brain tumor response. The median brain progression-free survival was 8.60 (95% CI [6.43–13.33]) months and the median overall survival was 16.8 (95% CI [13.3–27.7]) months. Three patients had in-field failure, five had out-field failure and two had in-field and out-field failure.

Conclusion

WBRT plus SIB led to improved tumor control and clinical outcome in breast cancer patients with brain metastasis.

Introduction

Breast cancer is the leading malignancy in women worldwide. It is estimated that 281,550 new cases of female breast cancers in United States, leading to around 43,600 death in 2021 (Siegel et al., 2021). In China, these numbers were 268,600 new cases and 69,500 deaths in 2015 (Chen et al., 2016). Breast cancer is a common causal disease of cancer brain metastasis. Up to 5.1% of all breast cancers would develop brain metastasis (BM) and 14.2% of metastatic breast cancer were presented with BM (Barnholtz-Sloan et al., 2004). BM is a significantly poor prognostic factor in breast cancer patients, with the median overall survival being only around four months (Castaneda et al., 2015). Associated poor prognostic factors include advanced stage, higher grade, estrogen receptor (ER) negativity, shorter time to develop BM and >3 brain lesions (Castaneda et al., 2015). Since the incidence of BM is increasing, it represents a critical problem that is not only life-threatening to breast cancer patients but also significantly increase the cost of medical care (Pelletier et al., 2008).

Treatments of BM from breast cancers have evolved with the progression of radiotherapy techniques, and the development of novel chemotherapeutic and targeted drugs. Surgery is often firstly recommended whenever eligible. Stereotactic radiosurgery (SRS) shows similar tumor control effect but reduces brain irradiation. Whole brain radiotherapy (WBRT) is now increasingly challenged given significant cognitive impairment, especially for those long-survived patients following targeted therapies (Chargari et al., 2010). While multiple disciplinary treatments are always encouraged for breast cancer patients presented with brain metastasis, the outcome of these patients remains dismal (Dawood et al., 2010; Jeene et al., 2018).

In addition to technical accessibility, SRS is generally used in highly selected patients such as with limited number (≤3) and relatively small sized (<3 cm) brain lesions, well controlled primary and other metastatic disease, high performance status score and younger age. Therefore, not all breast cancer patients with BM could benefit from SRS (Halasz et al., 2016; Possanzini & Greco, 2018). Alternatively, WBRT with boost irradiation to the brain lesions can also increase local dose whilst spare adjacent brain tissue from excessive irradiation and can be considered to treat BM when SRS is not feasible (Rodrigues et al., 2012; Yang et al., 2017; Zhong et al., 2020). Previous studies indicated that WBRT+SIB might lead to comparable tumor control while preserve cognitive functions (Bauman et al., 2016; Yang et al., 2017). However, most of these studies have enrolled BM from diverse primary cancers, especially from lung cancers (Du et al., 2021; Qing et al., 2020). The clinical outcome of BM from breast cancers treated with WBRT and simultaneous integrated boost (SIB) is not fully demonstrated.

In this study, we reported the clinical outcome and associated prognostic factors in breast cancer patients with brain metastasis treated with WBRT plus SIB using a simultaneous integrated boost by intensity modulated radiotherapy (SIB-IMRT) approach. We hope to provide valuable clues for future applications of WBRT plus SIB in these patients.

Materials and Methods

Patient cohort

We retrospectively reviewed consecutive breast cancer patients who developed brain metastasis and treated in our institute from March 2017 to January 2021. Inclusion criteria were: (1) age ≥ 18 years old; (2) female patients with pathologically confirmed breast invasive ductal carcinoma or invasive lobular carcinoma; (3) image or pathological diagnosed brain metastasis; (4) performance status score ≤ 2; (5) life expectancy ≥ 3 month; (6) adequate hematological and organ functions; (7) completed WBRT plus SIB to the brain lesions using SIB-IMRT approaches with or without systemic treatment. Exclusion criteria were: (1) severe comorbidities that hindered radiotherapy; (2) previous irradiation to the brain; (3) symptomatic cerebral infarction or cerebral hemorrhage; (4) coexistence of other malignancies; (5) pregnancy or lactation. This study was approved by the Medical Ethics Committee, Zhongnan Hospital of Wuhan University (No. 2022012). Informed consent was obtained from all included patients or their next of kin.

Radiotherapy

Twenty-seven patients were eligible for analysis. All patients received hypofractionated radiation therapy plus WBRT using SIB-IMRT approaches (Fig. 1). Specifically, 19 patients received 46.8 Gy in 13 fractions for brain lesions and 32.5 Gy in 13 fractions for whole brain; 4 patients received 36 Gy in 10 fractions for brain lesions and 30 Gy in 10 fractions for whole brain; one patient received 45 Gy in 15 fractions for brain lesions and 37.5 Gy in 15 fractions for whole brain; one patient received 50 Gy in 20 fractions for brain lesions and 40 Gy in 20 fractions for whole brain; one patient received 33 Gy in 11 fractions for whole brain and a boost of 16 Gy in four fractions for brain lesions; one patient received 30 Gy in 10 fractions for whole brain and a boost of 24 Gy in eight fractions for brain lesions. Dose constrains for critical organs at risk were listed in Table S1.

Figure 1 Dose distribution of WBRT plus SIB in a representative breast cancer patient with multiple brain metastases.

WBRT, whole brain radiotherapy; SIB, simultaneous integrated boost. Representative figures were shown, with the consent of publication of anonymous use of patients’ imaging.

Systemic treatment

Twenty-five patients received venous or oral chemotherapy after brain irradiation. Chemotherapeutic agents included vinorelbine, gemcitabine, capecitabine, cisplatin, paclitaxel, temozolomide. Sixteen patients received HER2-targeted therapy such as trastuzumab, pertuzumab, lapatinib, pyrotinib, TD-M1. Two patients received VEGFR-targeted therapy apatinib. Three patients received endocrine therapy including fulvestrish and anastrozole.

Statistical analysis

Categorical variables were described as frequencies (percentages) and were compared with Chi-square and Fishers’ exact test. Continuous variables were presented as median (range) and were analyzed with Mann-Whitney test. Kaplan-Meier plots were used to compare patient survival and log-rank test and Cox regression were used for survival analysis. R (4.0.3; R Core Team, 2020) and SPSS (Version 22.0. Armonk, NY: IBM Corp) software packages were used to perform statistical analysis. All tests were two-sided and for all statistical tests, a p < 0.05 was considered significant unless otherwise specified.

Results

Clinical characteristic of breast cancer patients with brain metastasis

From March 2017 to January 2021, twenty-seven female breast cancer patients presented with image or pathology diagnosed brain metastasis were included into analysis. The median age was 51 (range: 33–79) years old. Pathologically, most were invasive ductal carcinoma (26/27, 96.3%) and were WHO grade II (19/27, 70.4%) diseases. Hormone receptors (either estrogen receptor or progesterone receptor) were expressed in 51.9% (14/27) of patients. HER2 were overexpressed in 59.3% (16/27) of patients. There were four (14.8%) triple negative breast cancer (TNBC) patients. The median Ki-67 expression level was 35% (range: 5–80%). The median number of brain metastasis is 3 (range: 1–9). In addition, 12 (44.4%), 15 (55.6%), 7 (25.9%) and 10 (37.0%) patients were presented with extracranial bone, lung, liver and other metastasis, respectively (Table 1).

Table 1 Patient characteristics.

	Overall n = 27	
Age, median (range), years	51 (33–79)	
Comorbidity		
Diabetes	2 (7.4%)	
CHD	1 (3.7%)	
COPD	0	
Hypertension	0	
Pathology		
Invasive ductal carcinoma	26 (96.3%)	
Invasive lobular carcinoma	1 (3.7%)	
WHO grade		
I	1 (3.7%)	
II	19 (70.4%)	
III	7 (25.9%)	
ER		
Positive	14 (51.9%)	
Negative	12 (44.4%)	
Unknown	1 (3.7%)	
PR		
Positive	10 (37.0%)	
Negative	16 (59.3%)	
Unknown	1 (3.7%)	
HER2		
Positive	16 (59.3%)	
Negative	10 (37.0%)	
Unknown	1 (3.7%)	
TNBC		
Yes	4 (14.8%)	
No	22 (81.5%)	
Unknown	1 (3.7%)	
Ki-67, median (range), %	35 (5–80)	
Number of brain metastasis, median (range)	3 (1–9)	
Extracranial metastasis		
Bone metastasis	12 (44.4%)	
Lung metastasis	15 (55.6%)	
Liver metastasis	7 (25.9%)	
Other metastasis	10 (37.0%)	
Note:

CHD, coronary heart disease; COPD, chronic obstructive pulmonary disease; ER, estrogen receptor; HER2, human epidermal growth factor receptor 2; PR, progesterone receptor; TNBC, triple-negative breast cancer.

Clinical outcomes of breast cancer patients with brain metastasis

All patients received WBRT plus SIB. The median biologically effective dose (BED) of brain metastatic lesions was 63.65 (range: 48.96–70.20) Gy and the median BED of whole brain irradiation was 40.63 (range: 39.00–48.00) Gy. Twenty-five (92.6%) patients also received systemic chemotherapy. Fourteen of the 16 patients with HER2 positive breast cancers received systemic HER2-targeted treatment and 14 of them received anti-HER2 tyrosine kinase inhibitors such as lapatinib and pyrotinib. Two patients received VEGFR-targeted therapy apatinib. Three patients received endocrine therapy including fulvestrish and anastrozole. In total, four (14.8%) patients achieved clinical complete response of brain lesions and 14 (51.9%) were partial response. The other nine patients were not evaluated for brain tumor response. The median values of GTV volumes, maximum doses (Dmax) of GTV and mean doses (Dmean) of the brain in the entire cohort were 8.80 (interquartile range: 3.15–20.90) cm3, 4,963.80 (interquartile range: 4,915.58–5,039.60) cGy and 3,513.50 (interquartile range: 3,455.22–3,592.72) cGy, respectively. We did not observe significant differences of these values among patients with different response patterns (Table S2). After a median follow-up of 30.1 months, 19 patients developed brain lesion progression while the other eight patients were not evaluated. Eleven patients experienced neurological side effects including as reduced memory, dizziness, retarded reaction and apathy. Until the last follow up by June 30, 2021, 20 (74.1%) patients died and 15 were due to extracranial progression, while another five were due to intracranial progression (Table 2). The median brain progression-free survival was 8.60 (95% CI [6.43–13.33]) months and the median overall survival was 16.8 (95% CI [13.3–27.7]) months (Fig. 2).

Table 2 Treatments and patient outcomes.

	Overall n = 27	
Radiotherapy doses		
BED of GTV (Gy), median (range)	63.65 (48.96–70.20)	
BED of WBRT (Gy), median (range)	40.63 (39.00–48.00)	
Systemic treatment		
Chemotherapy	25 (92.6%)	
HER2-targeted therapy	16 (66.7%)	
VEGFR-2 targeted therapy	2 (7.4%)	
Endocrine therapy	3 (11.1%)	
TKI treatment for HER2+ (n = 16)		
Yes	14 (87.5%)	
No	2 (12.5%)	
Brain tumor response		
Complete response	4 (14.8%)	
Partial response	14 (51.9%)	
Unevaluated	9 (33.3%)	
Neurological side effects	11 (40.7%)	
Brain progression		
Yes	19 (70.4%)	
Unevaluated	8 (29.6%)	
Death		
Yes	20 (74.1%)	
No	7 (25.9%)	
Cause of death (n = 20)		
Extracranial progression	15 (75.0%)	
Intracranial progression	5 (18.5%)	
Note:

BED, biologically effective dose; GTV, gross tumor volume; HER2, human epidermal growth factor receptor 2; TKI, tyrosine kinase inhibitor; VEGFR-2, vascular endothelial growth factor-2; WBRT, whole brain radiotherapy.

Figure 2 Intracranial progression-free survival and overall survival of breast cancer patient with multiple brain metastases.

PFS, progression-free survival; OS, overall survival.

Prognostic factors associated with patient survival

We then compared overall survival of patients with different clinical features. We found that HER2-positve patients showed improved survival as compared to HER2-negative ones (21.3 vs. 9.17 months, P = 0.042) (Fig. 3A). However, there was no difference between hormone receptor positive and negative patients, nor between TNBC and non-TNBC patients (Figs. 3B, 3C). Regarding the number of brain metastasis, we did not find significant difference between patients with ≤3 and those with >3 lesions (Fig. 3D). In terms of extracranial metastasis, both liver (13.4 vs. 20.4, P = 0.049) and bone metastasis liver (13.3 vs. 20.4, P = 0.041) were linked with reduced overall survival (Figs. 3E, 3F). Mean dose of brain did not significantly impact on patients’ overall survival (Fig. S1). To explore potential prognostic factors that were associated with patient survival, we performed univariate analysis with Cox proportional model (Table 3). We observed that the expression of progesterone receptor (PR) was associated with poor prognosis (HR = 4.79, 95% CI [1.29–17.9], P = 0.0196) while the expression of HER2 tended to be associated with improved overall survival (HR = 0.35, 95% CI [0.12–1.01], P = 0.0512). In addition, liver and bone metastasis showed trend to be associated with increased risk of death, although both were marginally statistically significant. Other factors including age (>50), ER expression, grade (WHO III), number of brain lesion (>3), Ki-67 expression (>35%), lung metastasis, and other metastasis were not significantly associated with patients’ overall survival (Table 3). However, none of these factors showed to be significantly associated with patients’ prognosis upon multivariate Cox analysis (Table 3).

Figure 3 Overall survival of patients by their clinical features.

Patient survival by (A) HER2 expression status, (B) HR expression status, (C) TNBC, (D) number of brain lesions, (E) liver metastasis, and (F) bone metastasis. HER2, human epidermal growth factor receptor 2; HR, hormonal receptor; TNBC, triple-negative breast cancer.

Table 3 Univariate and multivatiate analysis of overall survival.

	Univariate analysis	Multivariate analysis	
	HR (95% CI)	P value	HR (95% CI)	P value	
Age (>50)	0.63 [0.24–1.62]	0.335	0.90 [0.148–5.5]	0.91	
ER (positive)	1.15 [0.44–3.01]	0.77	0.50 [0.052–4.8]	0.549	
PR (positive)	4.79 [1.29–17.9]	0.0196	26.73 [0.369–1,937.8]	0.133	
HER2 (positive)	0.35 [0.12–1.01]	0.0512	0.66 [0.030–14.7]	0.794	
TNBC (yes)	2.13 [0.67–6.82]	0.202	5.43 [0.4274–69.4]	0.193	
Grade (III)	1.54 [0.57–4.18]	0.399	1.82 [0.433–7.7]	0.414	
Number (>3)	0.89 [0.32–2.54]	0.834	0.62 [0.071–5.4]	0.668	
Ki-67 (>35%)	0.89 [0.35–2.26]	0.805	0.50 [0.094–2.7]	0.418	
Lung metastasis (yes)	0.85 [0.33–2.16]	0.729	3.53 [0.526–23.8]	0.194	
Liver metastasis (yes)	2.72 [0.97–7.66]	0.0583	5.48 [0.802–37.5]	0.083	
Bone metastasis (yes)	2.93 [0.99–8.58]	0.0505	1.04 [0.170–6.4]	0.962	
Other metastasis (yes)	2.07 [0.76–5.66]	0.157	5.41 [0.887–32.9]	0.067	
Note:

ER, estrogen receptor; HER2, human epidermal growth factor receptor 2; HR, hazard ratio; PR, progesterone receptor; TNBC, triple-negative breast cancer.

Patterns of brain recurrence or progression

Ten patients were eligible to be evaluated for brain disease recurrence of progression. Three patients had in-field recurrence or progression, five had out-field recurrence or progression, and two patients had both in- and out-field recurrence or progression. Representative patients with three different patterns of brain recurrence or progression were shown in Fig. 4.

Figure 4 Representative cases of brain lesion recurrence/progression.

(A) Intra-field recurrence/progression, (B) out-field recurrence/progression, (C) mixed recurrence/progression. Red arrows indicated intra-filed brain lesions and yellow arrows indicated out-field brain lesions. Representative figures were shown, with the consent of publication of anonymous use of patients’ imaging.

Discussion

In this study, we demonstrated that WBRT plus SIB had promising effect on intra-cranial tumor control and could improve patient survival in breast cancer with brain metastasis, especially in patients with HER2-postive tumors. This treatment approach could be recommended for patients with multiple brain metastases when SRS is not eligible.

Brain metastases are common in late-stage lung cancers, breast cancers and malignant melanoma (Lorger & Felding-Habermann, 2010). With a significantly prolonged overall survival time of breast cancer patients and the effective control of tumor recurrence or metastasis to other parts of the body by chemotherapy and targeted drugs, the incidence of breast cancer brain metastasis is increasing steadily (Lombardi et al., 2014). Risk factors of developing BM in breast cancer patients included younger age, coexisting pulmonary or liver metastasis, hormone receptor negativity, HER2 positivity, more metastatic lesions, and larger BM lesion size (Aoyama, 2011). The probability of brain metastases from breast cancer with obvious clinical symptoms is 10% to 15%, and the prevalence could be as high as 30% if asymptomatic brain metastases were taken into account, putting forward a clinical problem that needs to be solved urgently (Lin et al., 2013).

Although systemic chemotherapy for breast cancer brain metastasis has made substantial progress in recent years, local therapies including surgical resection, whole brain radiation therapy (WBRT) and stereotactic radiotherapy (SRS) remain current first-line treatment for brain metastasis from breast cancer. BM from breast cancer with limited number (≤3–4 lesions) and small size was mainly treated with SRS. The dose of SRS ranged from 16 to 20 Gy (median 18 Gy). The median cranial PFS was 5.7 months and the median OS was 9.8 months (Kelly et al., 2012). Patients treated with SRS with or without WBRT had significantly higher overall survival than those treated with WBRT alone (Andrews et al., 2004). Nevertheless, in cases of more brain lesions (>4 lesions) and larger tumors and in other situations where SRS is not suitable, WBRT remain major treatment approach.

Recent studies have shed light on the value of WBRT plus SIB that can achieve high dose to the brain lesions whilst protect normal brain tissue from exceeded irradiation. Similar to SRS, SIB using modern radiotherapy technology such as intensity-modulated radiotherapy, helical tomotherapy and imaging-guided radiotherapy could increase local irradiation dose to the brain lesions whilst deliver relative low dose to the adjacent brain tissue. This approach has been shown to improve brain tumor control, reduce the appearance of new lesions, and protect the quality of life of cancer patients. In brain metastasis from lung cancer, WBRT+SIB yielded improved intracranial PFS, reduced risk of progression outside tumor irradiation field as compared with WBRT + SRS, while the response rates, overall survival and toxicity were similar between the two treatment modalities (Lin et al., 2021). In a phase II study of hippocampal-sparing WBRT + SIB in brain metastasis, 20 Gy in 10 fractions were delivered to the whole brain plus a SIB of 40 Gy in 10 fractions was delivered to the metastatic brain lesions. Median PFS was 2.9 months and median OS was 9 months. Note that the majority primary cancers were lung cancers, and only 10% were breast cancers (Westover et al., 2020). It was suggested that hippocampal sparing WBRT + SIB allowed the preservation of cognitive functions while obtaining favorable cancer control.

Our study showed that WBRT plus SIB was feasible in patients with multiple BM from breast cancer. The number of lesions could be as large as nine, which was generally not amenable for SRS. The median overall survival was 16.8 months, which was comparable with or superior over published data (Chen et al., 2020; Popp et al., 2020; Westover et al., 2020). This could be explained by the fact that in other series, more patients were with metastatic lung cancers that were of poorer prognosis. In addition, in our study, the biologically effective dose given to the lesions was higher (median: 63.65 Gy, range: 48.96–70.20 Gy). Overall, the treatment was well tolerated, with 11 (40.7%) patients developed neurological side effects, mostly presented as reduced memory, dizziness, retarded reaction and apathy.

Our results also showed a promising intracranial control with WBRT plus SIB. Four (14.8%) patients achieved clinical complete response and 14 (51.9%) had partial response. The median progression-free survival was 8.60 months with at least 19 (70.4%) patients eventually developed intracranial recurrence or progression. In our series, 25 patients had extracranial metastasis and for those 20 patients died, 15 (75.0%) died because of extracranial progression whereas only five (25.0%) were due to intracranial progression. We documented both intra- and extra-field recurrence or progression in these patients, suggesting that more efficient systemic treatment is needed to improve treatment outcome. We expected that these encouraging results could be validated in further studies. Indeed, a prospective phase II clinical study that investigate the effect and cognition preservation of hippocampal-avoidance WBRT (HA-WBRT) plus SIB as compared with HA-WBRT alone is ongoing (NCT04452084) (Chia et al., 2020).

From univariate analysis of this series of patients, we found that neither high number of brain lesions (>3), tumor grade, nor ER expression was associated with overall survival. On the other hand, and perhaps not surprisingly, liver (P = 0.0583) and bone metastasis (P = 0.0505) were marginally associated with reduced overall survival. In addition, we observed a marginal significant improved prognosis in HER2 positive patients (HR = 0.35, 95% CI [0.12–1.01], P = 0.0512). This could be due to the use of anti-HER2 treatment, especially with the small kinase inhibitors such as lapatinib that could pass through blood-brain barrier (Christodoulou et al., 2017). There were 16 HER2 positive patients and 14 (87.5%) of them received anti-HER2 treatment. In other studies, similar results were also reported. In a study of 432 HER2 positive breast cancers with BM, 37.5% were ER positive HER2 positive and 62.5% were ER negative HER2 positive, and the median overall survival was 16.5 months and 11.5 months, respectively. Treatment with trastuzumab and lapatinib was associated with improved survival as compared with trastuzumab or lapatinib alone treatment or no HER-2 targeted treatment (Hayashi et al., 2015). HER2 negativity and progressive extracranial disease were negative prognostic factors (Kelly et al., 2012). Notably, the incidence of brain metastases in triple-negative breast cancer is 25–46%, and the time of brain metastasis is significantly earlier than that of HER2-positive breast cancer and hormone receptor-positive breast cancer (Jorns, Healy & Zhao, 2013). In the absence of effective targeted treatment, the treatment of TNBC patients with brain metastasis remains one particular challenge.

We did not detect the impact of ER expression on patient survival, and we even saw a reduced prognosis in PR positive patients. The prognostic role of hormone receptor in breast cancer patients with brain metastasis remains controversial. Some showed that patients with hormone receptor-positive breast cancer with brain metastases have a better prognosis, with a median OS of 15–17 months (Buonomo et al., 2017). Others ER expression did not significantly impact on overall survival (Mix et al., 2016). Our results demand to be validated in larger studies.

There were several limitations in our study. First, this was a single-center retrospective study with limited number of patients. There was inevitable selection and information bias. Second, this was a single arm study. Therefore, any comparative analysis should be interpreted with caution. Third, there was no information about hippocampal avoidance and no evaluation of cognitive function. While the combination of whole-brain radiation therapy (WBRT) with stereotactic body radiation therapy (SBRT) or stereotactic radiosurgery (SRS) has been extensively studied, our research focused on the combination of WBRT with SIB, which is a relatively new area of exploration. Our study indicated that the combination of WBRT and SIB not only showed good safety but also led to potential efficacy, providing a viable treatment option for brain metastasis in breast cancers. We believe that despite the retrospective nature of our study, its findings have clinical implications. The treatment regimen of WBRT combined with SIB is easy to implement and cost-effective, which is particularly important for medical institutions that cannot readily implement SBRT or SRS.

In conclusion, in this retrospective study, we demonstrated that WBRT plus SIB was feasible, well tolerated, and led to improved tumor control and patient outcome in breast cancer patients with brain metastasis. Prospective comparative studies are needed to confirm the efficacy of this radiotherapy strategy.

Supplemental Information

Supplemental Information 1 Raw data.

The patient characteristic and treatment outcomes and univariate analysis of overall survival.

Supplemental Information 2 Overall survival of patients by the mean dose of brain.

Supplemental Information 3 Dose constraint of organs at risk.

Supplemental Information 4 Comparison of radiation volumes and doses.

We would like to acknowledge all the participants for their participation and their contribution to this study.

Additional Information and Declarations

Competing Interests

Author Contributions

Human Ethics

Data Availability

The authors declare that they have no competing interests.

Hongyan Zhang conceived and designed the experiments, performed the experiments, analyzed the data, prepared figures and/or tables, authored or reviewed drafts of the article, and approved the final draft.

Qiuji Wu conceived and designed the experiments, performed the experiments, analyzed the data, prepared figures and/or tables, authored or reviewed drafts of the article, and approved the final draft.

Li Li conceived and designed the experiments, performed the experiments, authored or reviewed drafts of the article, and approved the final draft.

Linwei Wang conceived and designed the experiments, performed the experiments, authored or reviewed drafts of the article, and approved the final draft.

Yahua Zhong conceived and designed the experiments, prepared figures and/or tables, authored or reviewed drafts of the article, and approved the final draft.

The following information was supplied relating to ethical approvals (i.e., approving body and any reference numbers):

This study was approved by the Medical Ethics Committee, Zhongnan Hospital of Wuhan University (No. 2022012).

The following information was supplied regarding data availability:

The raw data are available in the Supplemental Files.

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
