# Peer review of "Whole-brain radiation therapy plus simultaneous integrated boost for brain metastases from breast cancers"

_PeerJ, doi:10.7717/peerj.17696_

## Round 0.1 · original submission · Major Revisions

Dear authors, many thanks for your submission. Upon peer-reviewing i decided that your manuscript does require a few revisions before it can be published. Please, refer to the reviewers' comments. Also, be mindful of revision language and formatting. Include succinct but comprehensive legends in your figures or tables. Many thanks.

**Language Note:** The Academic Editor has identified that the English language must be improved. PeerJ can provide language editing services - please contact us at [email protected] for pricing (be sure to provide your manuscript number and title). Alternatively, you should make your own arrangements to improve the language quality and provide details in your response letter. – PeerJ Staff

Reviewer 1 ·

Basic reporting

■I think that sentence is well written.
But there are some problem about next step

Experimental design

■I think that there are next problem for this study.
①RPA/GPA
→What is the RTOG-RPA and GPA? I think that RPA and GPA are prognostic factor.
②Waht is brain tumor volume?
→Is there a relationship between the tumor volume and tumor response?
③whole brain radiation dose
→The whole brain radiation doses are heterogeneous. Is it possible that these heterogeneous radiation dose could have affected survival rates?
④IMRT
→Is IMRT technique VMAT? step and shoot? Tomotherapy?
→Is the IMRT radiation dose D95? D50? isodose line prescription?
→What is the max dose in the tumor?
→WHat is the dose limitation?(brain, brain stem, chiasm, eye, optic nerve,・・)
if the radiation dose prescriptions methods were different, I think this differentiation cause vias.
⑤Radiation method
→please write the linac and RTP name.
→Did you use same Linac?What is the MV and the range of MLC?
→What is the Name of RTP? What is the calculation algorithm?
→What is the Thickness of slice at planning CT?
→Did you use CT/MRI fusion? What is the name of MRI? please write Tesla)
→Did you use Gd enhancement MRI?
→Did you use IGRT at radiation?
 ⑥chemotherapy
→After brain radiation therapy, what was the chemotherapy situation?
⑦The results
→Why did you use only univariate analysis for OS? I think that multivariate analysis is necessary.
⑧CR/PR/PD
→IS the method of evaluation of CR/SD/PR are suitable? especially timing/
⑨abstract
→Method: Four patients . achieved・・→→write at results. Please check and correct the method.
at abstract
⑩ neues
→What is the neues?
→In the next article:Survival time and prognostic factors after whole-brain radiotherapy of brain metastases from of breast cancer the radiation dose (high dose) and HER2 type are prognostic factor
in the breast cancer patients with new brain metastases.

Validity of the findings

please check the comments at Experimental design.

Additional comments

I want to read the article again after revised.

Reviewer 2 ·

Basic reporting

The number of patients is low and it is a non-homogeneous study. However, I find it important because it is one of the few studies in which SIB and SBRT were conducted simultaneously and in a patient group with a median survival of 16 months.

Experimental design

Experimental design is good.

Validity of the findings

Generally good, but some minor corrections were suggested in the article and in the table

Additional comments

1. Although whole brain daily fractions are compatible with the literature, similar studies by Aoyama and I have also shown that levels above 250 cGy can be toxic and should be taken into consideration in future studies.
2.Total brain doses are also high. However, since the results were good, some new modalities such as SBRT SIB may have contributed here, regardless of toxicity. Toxicity rates in patients with WBRT doses above BED 4000 cGy should be reported separately.
3.Grade I, II and III toxicity rates should be reported separately according to RT doses.
4.In addition to performance scoring information, changes before and after RT should be reported.
5. While the response rates were given, it was reported that relapses were more common outside the field. Since there is a field in WBRT, the term out field should be corrected to out field of SBRT. It should be noted that recurrence rates are lower or higher in the dose range of WBRT. Most of the patients were prescribed SBRT even though a hypofractionated regimen was applied as SIB. It should be corrected by SBRT or hypofractionated RT.
6. A space should be added to the words to the left of the reference information in parentheses in the text and to those that do not have a space between the word before the parentheses.
7. In some parts of the text, HER2 positive or negative is written, and in others HER2+ or – is written. The standard HER2 positive or negative should be written in all of them.
8. In Table 1, although the right and left sides of the table are independent from each other, they are combined in the same table. It was wrong. It should be rearranged in a comprehensible way as 2 separate tables.

Annotated reviews are not available for download in order to protect the identity of reviewers who chose to remain anonymous.

---

## Round 0.2 · Major Revisions

Dear authors, unfortunately the reviewers opinions divided this time. I decided to provide you with extra time for revisions in order for it to be a relevant piece of literature. The main concerns are centred in sample size being small and the conclusion that HER2 positive group show good prognosis being widely known.
Indeed, a quick search in PubMed return papers like: doi: 10.3389/fonc.2022.903001, https://doi.org/10.1038/s41598-023-38200-y , https://doi.org/10.1186/s40001-022-00894-7

PeerJ does not require novelty for publishing a scientific body of work. However, I do recommend that you address these core situations properly and clearly in your manuscript in order to also justify the efforts you have put into this research.

Reviewer 1 ·

Basic reporting

Thank your submission. I thnk this rexearch is important.But the conclusion is not new.The conclusion that HER2 positive group show good prognosis is widely knowen.

Experimental design

I think that multivariate analysis is essential.

Validity of the findings

.The conclusion that HER2 positive group show good prognosis is widely knowen.

Reviewer 2 ·

Basic reporting

This article may make an important contribution to the literature for SIB RT in brain metastases of breast cancer.

Experimental design

Good

Validity of the findings

Good

Additional comments

Accepted for publishing

---

## Round 0.3 · accepted · Accept

Dear authors,

i am now accepting your work for publication in PeerJ. It is my current belief that your transparent work and manuscript addresses properly the limitations previously highlighted by the reviewers. Many congratulations. And thank you.